# Conserved Role of Heterotrimeric G Proteins in Plant Defense and Cell Death Progression

**DOI:** 10.3390/genes15010115

**Published:** 2024-01-18

**Authors:** Parastoo Karimian, Yuri Trusov, Jose Ramon Botella

**Affiliations:** School of Agriculture and Food Sciences, University of Queensland, St. Lucia, Brisbane, QLD 4072, Australia; prstkrmn@gmail.com (P.K.); y.trusov@uq.edu.au (Y.T.)

**Keywords:** programmed cell death, heterotrimeric G proteins, plant immunity, guard hypothesis

## Abstract

Programmed cell death (PCD) is a critical process in plant immunity, enabling the targeted elimination of infected cells to prevent the spread of pathogens. The tight regulation of PCD within plant cells is well-documented; however, specific mechanisms remain elusive or controversial. Heterotrimeric G proteins are multifunctional signaling elements consisting of three distinct subunits, Gα, Gβ, and Gγ. In *Arabidopsis*, the Gβγ dimer serves as a positive regulator of plant defense. Conversely, in species such as rice, maize, cotton, and tomato, mutants deficient in Gβ exhibit constitutively active defense responses, suggesting a contrasting negative role for Gβ in defense mechanisms within these plants. Using a transient overexpression approach in addition to knockout mutants, we observed that Gβγ enhanced cell death progression and elevated the accumulation of reactive oxygen species in a similar manner across *Arabidopsis*, tomato, and *Nicotiana benthamiana*, suggesting a conserved G protein role in PCD regulation among diverse plant species. The enhancement of PCD progression was cooperatively regulated by Gβγ and one Gα, XLG2. We hypothesize that G proteins participate in two distinct mechanisms regulating the initiation and progression of PCD in plants. We speculate that G proteins may act as guardees, the absence of which triggers PCD. However, in *Arabidopsis*, this G protein guarding mechanism appears to have been lost in the course of evolution.

## 1. Introduction

Programmed cell death (PCD) is a genetically regulated process essential for diverse developmental pathways, stress responses, cellular homeostasis, and the eradication of infected cells within plants [1]. PCD plays a significant regulatory role in plant immunity. Conventionally, it is postulated that PCD restricts pathogen proliferation exclusively to infection sites and precludes its dissemination to adjacent cells [2,3,4]. Given the potential deleterious effects of PCD on host tissues, its regulation is tightly controlled, encompassing at least three coordinated phases: initiation, progression or execution, and termination [5]. During initiation, cells receive specific signals activating intracellular pathways, starting the PCD cascade. The duration of PCD progression can vary, influenced by a combination of genetic and environmental factors. The termination of PCD entails the cessation of the PCD signaling cascade, the clearance of cellular debris generated during PCD, and the restoration of homeostasis within the surrounding tissue. While the initiation and termination stages of PCD have been relatively well-studied with established mechanisms, the progression phase remains elusive [4,5,6,7].

Heterotrimeric G proteins, hereinafter G proteins, composed of the Gα, Gβ, and Gγ subunits, are signaling complexes that play a role in a variety of physiological processes in eukaryotic cells. In animals, the activation of a G protein-coupled receptor by a ligand leads to the binding of guanosine triphosphate to the Gα subunit and its dissociation from the Gβγ dimer. Both units then transmit signals to downstream effectors, resulting in a range of cellular responses [8]. In plants, however, G proteins can function in a nucleotide-independent manner and without dissociation [9,10,11,12]. Although it was initially thought that the Gβγ dimer only played a supportive role for Gα, it is now well-established that the Gβ and Gγ subunits have their own signaling activities [13]. The Gβγ dimer can reversibly translocate from the plasma membrane to internal membranes and directly activate or inhibit specific downstream effectors [13,14,15,16,17]. In animals, the Gβγ dimer is now recognized as an important regulator of multiple cellular responses [14,16]. By contrast, the role of the Gβγ dimer in plants is not fully understood, and its molecular mechanisms in plant cells remain obscure.

Plant G proteins are involved in a wide range of cellular processes, including innate immunity and multiple defense-related responses [18,19,20,21,22,23,24,25,26]. Plants have a limited repertoire of G proteins compared to animals. In *Arabidopsis*, there are four Gα subunits (AtGPA1, XLG1, XLG2, and XLG3), one Gβ subunit (AGB1), and three Gγ subunits (AGG1, AGG2, and AGG3) [27,28,29,30,31,32]. XLG2 and XLG3, in combination with two Gβγ dimers (ABG1/AGG1 and AGB1/AGG2), provide resistance to a variety of pathogens, including viruses, bacteria, and fungi [18,22,33,34,35,36].

Several studies have established a positive connection between G proteins and PCD in plants. Specifically, XLG2 and AGGs are known to interact directly with transmembrane receptor-like kinases (RLKs), including BAK1-interacting RLK 1 (BIR1) and flagellin-sensitive 2 (FLS2) [10,33,37,38]. *Arabidopsis* mutants deficient in G proteins, such as *agb1*, *agg1 agg2*, and *xlg2 xlg3*, demonstrate markedly reduced reactive oxygen species (ROS) production induced by flagellin peptide flg22 [10,33,39,40]. Moreover, genetic studies involving these mutants suggest that G proteins contribute to the *bir1* autoimmune phenotype by sustaining a continuously activated PCD response [35,40,41]. In rice, the simultaneous overexpression of Gβ and Gγ subunits has been shown to enhance resistance to blight disease and abiotic stresses [25]. Collectively, these findings underscore the significant positive role of G proteins, particularly Gβγ, in regulating PCD processes in plants.

In contrast to the findings from *Arabidopsis*, studies involving Gβ knockouts in rice, maize, tomato, and cotton, along with the triple XLG knockout in maize, reveal severe autoimmune symptoms. These symptoms comprise heightened PR gene expression, elevated ROS production, and increased seedling lethality [19,42,43,44]. Such observations suggest that, in these contexts, G proteins might exert a negative influence on PCD processes; however, the underlying mechanism is not established.

In this study, our objective was to explore the role of G proteins in PCD using an overexpression approach, complementing the insights gained from knockout mutations. Our findings indicate that G proteins from both *Arabidopsis* and tomato promote the progression of PCD, rather than its initiation, across evolutionarily diverse eudicot species such as *Arabidopsis*, tomato, and *N. benthamiana*. Additionally, our results demonstrate that the AGB1/AGG1 and XLG2 subunits cooperatively contribute to this enhancement in a mutually supportive manner. Based on our findings and previous reports, we hypothesize that G proteins play a dual role in regulating PCD in plants. Specifically, these proteins exert a positive influence on the progression of PCD in all tested plants. Their apparent negative effect could potentially be explained within the framework of the guard hypothesis, suggesting that certain G protein subunits serve as guardees. Consequently, their absence triggers the initiation of PCD in the cell. However, in *Arabidopsis*, it seems that the guarding mechanism based on G proteins has been lost during evolution.

## 2. Materials and Methods

### 2.1. Plant Growth Condition

All mutants used in this study have the ecotype Columbia-0 (Col-0) background and are previously described, as follows: *agb1-2* [23], *gpa1-3* [45], *xlg2-1* [27], *xlg3* [46], *xlg2 xlg3* [27], *xlg1 xlg2 xlg3* [47], *NahG* [48], and *pad4* [49]. *Arabidopsis* plants were grown in a growth chamber under short-day conditions (8 h light/16 h dark) at 23 °C. *N. benthamiana* and *S. lycopersicum* seeds were grown under long-day conditions (16 h light/8 h dark) at 23 °C.

### 2.2. Transient Overexpression Vectors and Infiltration

The pCambia1300 with HiBiT-YFP-AGB1, -AGG1, and -XLG2 were described previously [9,10]. The geminiviral pBYR2eFa vector based on the bean yellow dwarf virus has been previously described [50]. Full length cDNA of *A. thaliana* AGB1, AGG1, and XLG2 and *S. lycopersicum* SlGB1 and SlGGA1 were cloned in pBYR2eFa vector and introduced into *Agrobacterium tumefaciens*, GV3101 strain. Primers for cDNA amplification are shown in Appendix A.

Cultures were grown in Luria–Bertani (LB) broth overnight at 30 °C with shaking at 250 RPM. The next day, the cultures were washed twice with 10 mM MgCl_2_, re-suspended in the infiltration buffer (10 mM MgCl_2_ and 100 μM acetosyringone (Sigma, Burlington, MA, USA)), and incubated at room temperature for four hours before infiltration. The concentration of the cultures determined at OD 600 was adjusted to 0.4 for *Arabidopsis* and 0.02 for *N. benthamiana*. Leaves of five-week-old plants, either *Arabidopsis* or *N. benthamiana*, were infiltrated with a needleless syringe.

### 2.3. RT-qPCR

Total RNA was isolated from leaves using a Maxwell^®^ RSC Plant RNA Kit (Promega, Madison, WI, USA) with a Maxwell^®^ RSC instrument. cDNA was synthesized using an iScript TM cDNA Synthesis Kit (Bio-Rad, Hercules, CA, USA) according to the manufacturer’s protocol. The RT-qPCR was conducted on a Light Cycler 96 system using FastStart SYBR Green Master (Rocher, Indianapolis, IN, USA) according to the manufacturer’s protocol. Relative expression levels of the tested genes were normalized with the reference gene *SAND* (AT2G28390). PR gene identifiers were as follows: *PR1* (AT2G14610), *PR2* (AT3G57260), *PR3* (AT3G12500), *PR4* (AT3G04720), and *PR5* (AT1G75040). RT-qPCR primers are listed in Appendix A.

### 2.4. Trypan Blue Staining

Infiltrated leaves were boiled in lactophenol–trypan blue solution (10 mL lactic acid, 10 mL phenol, 10 mL water, 20 mg trypan blue, and 80 mL ethanol) for two minutes and incubated on a rocking platform at room temperature overnight. Next, leaves were washed with chloral hydrate solution (2.5 g/mL water) for 3–5 h and photographed.

### 2.5. DAB Staining

Leaves were placed in 3′3-diaminobenzidine (DAB) solution (50 mg of 3′3-diaminobenzidine tetrahydrochloride hydrate (Sigma-D5637) in 47.5 mL water, pH 3.0, 0.05% Tween 20, and 2.5 mL of 200 mM Na_2_HPO_4_) and subjected to vacuum for 20 min, then incubated at room temperature overnight. The next day, the samples were boiled in the bleaching solution (3:1:1 ethanol:acetic acid:glycerol) for 15 min. Samples were preserved in a 1:4 glycerol:ethanol solution and photographed.

### 2.6. flg22-Induced ROS

*Arabidopsis* and *N. benthamiana* leaves infiltrated with GV3101 *A. tumefaciens* were collected 3 days after infiltration and placed in a 96-well plate. Following the addition of 150 μL distilled water to each well, the plate was kept in a dark cabinet overnight. The water was replaced with 100 μL of the reaction solution (peroxidase and luminol, 200 μg/mL each with 1 μM flg22). Luminescence was measured in a GloMax 96 Microplate Luminometer (Promega) [35].

### 2.7. Electrolyte Leakage Assay

Discs from *Arabidopsis* and *N. benthamiana* transiently expressing genes of interest were assayed 3 days after infiltration for electrolyte leakage with the electrolytic conductivity meter according to the protocol described previously [51].

### 2.8. Protein Content Assay

The 11-amino-acid peptide HiBiT [52] was fused to AGB1, AGG1, and XLG2, cloned into geminiviral vector pBYR2eFa, and transiently expressed in *N. benthamiana* leaves as explained above. To quantify protein levels, the leaf discs (6 mm diameter) were collected 3 days after infiltration and placed in a 96-well plate. This was followed by the addition of 30 μL of Nano-Glo^®^ HiBiT reagent (Promega). The light emission was evaluated based on relative light units determined using a GloMax^®^ Reader (Promega). The protein amounts positively correlated with relative light values (RLUs) according to the manufacturer’s protocol (Promega), making it possible to compare relative levels of expressed proteins by comparing corresponding RLUs.

### 2.9. Statistical Analyses and Software

Statistical analysis for two data sets was performed using unpaired Student’s *t*-test with two-tail distribution (heteroscedastic variance). A non-parametric Kruskal–Wallis test, followed by Dunn’s test for multiple pairwise comparisons was used for data which failed normality tests. The RT-qPCR results underwent normality testing using the Shapiro–Wilk test and were subsequently analyzed using parametric one-way ANOVA with Tukey’s multiple comparison test. Tukey’s test was employed as recommended for small sample sizes. A statistically significant difference was declared in all instances when the *p*-value was less than 0.05 (*p* < 0.05). To generate graphs and perform statistics analyses, GraphPad Prism 9.5.1.733 software was used.

## 3. Results

### 3.1. The Simultaneous Co-Expression of Gβ and Gγ Subunits Significantly Contributed to Cell Death Progression

In heterotrimeric G proteins, Gβ forms an obligate dimer with Gγ; when unbound, it becomes unstable. Research has indicated that the AGB1 protein levels are minimal in *Arabidopsis*
*agg1 agg2 agg3* triple mutants that lack all three Gγ subunits [53]. To confirm the requirement for Gγ, we quantified the relative levels of HiBiT-AGB1 fusion protein with and without co-expression of AGG1 using HiBiT-tag-based analysis. Our results showed that co-expression of AGB1 and AGG1 led to a 25-fold increase in AGB1 protein levels compared to expression of AGB1 alone (Figure 1A). Since Gβ and Gγ form a functional dimer [16], co-expression of both subunits is necessary for functional analyses.

Upon defense responses, programmed cell death often manifested as necrotic lesions. However, overexpression of AGB1 and AGG1 in *N. benthamiana* leaves rarely resulted in necrosis, despite achieving high expression levels. Surprisingly, we observed that infiltration of AGB1 and AGG1 significantly enhanced necrosis in older leaves that were already showing signs of senescence before infiltration. This observation prompted us to hypothesize that AGB1/AGG1 may function as an enhancer of cell death rather than a trigger. To test this hypothesis, we utilized the bean yellow dwarf viral (BeYDV) pBYR2eFa expression system, which induces mild necrosis in the host plant, possibly due to the expression of the viral Rep protein [50]. Co-expression of AGB1 and AGG1 using pBYR2eFa (pBYR2eFa-AGB1; pBYR2eFa-AGG1) showed enhanced necrosis in infiltrated tissues compared to the expression of GFP (pBYR2e-GFP) (Figure 1B). By contrast, necrosis was not observed in control infiltrations expressing AGB1/AGG1 or YFP using the pCambia1300 vector (Figure 1B). To confirm that AGB1/AGG1 promoted the PCD initiated by pBYR2eFa, we infiltrated leaves with three constructs: pCambia1300-AGB1, pCambia1300-AGG1, and pBYR2e-GFP. In this case, the lesions were similar to those produced via co-expression of pBYR2eFa-AGB1 and pBYR2eFa-AGG1 (Figure 1C), supporting our hypothesis that AGB1/AGG1 enhances cell death. The pBYR2eFa expression system was used in all subsequent experiments.

To further study the effect of AGB1/AGG1-overexpression in different plant species, transient assays were performed in *Arabidopsis* and *N. benthamiana* expressing pBYR2eFa-AGB1/pBYR2eFa-AGG1 or pBYR2e-GFP and tomato (*Solanum esculentum*) expressing pBYR2eFa-SlGB1/pBYR2eFa-SlGGA1 or pBYR2e-GFP. Infiltration of five-week-old *Arabidopsis* plants revealed that GFP overexpression caused only minor leaf discoloration, while overexpression of AGB1/AGG1 resulted in widespread necrosis (Figure 1D). To confirm the occurrence of PCD, we performed trypan blue staining, which showed intense staining in the area infiltrated with pBYR2eFa-AGB1/pBYR2eFa-AGG1, while GFP overexpression resulted in low staining levels (Figure 1D). Additionally, 3,3’-diaminobenzidine (DAB) staining revealed elevated levels of ROS in tissues overexpressing AGB1/AGG1, while tissues overexpressing GFP showed only minor brown coloration (Figure 1D). Similarly, overexpression of AGB1/AGG1 in *N. benthamiana* leaves resulted in strong trypan blue and DAB staining, while overexpression of GFP was associated with minor to no symptoms (Figure 1E). It is important to note that these results were not specific for the *Arabidopsis* AGB1 and AGG1 subunits. When the tomato SlGB1 and SlGGA1 were co-expressed in tomato leaves, stronger necrotic symptoms and stronger trypan blue and DAB staining were observed compared to pBYR2e-GFP expression (Figure 1F), indicating that the phenotypes do not depend on the source of the Gβγ dimer or the species being assayed. These results offer qualitative evidence suggesting that Gβγ contributes to the progression of PCD rather than its initiation across three different species.

### 3.2. Co-Expression of Gβγ Subunits Enhances flg22-Induced ROS Production and Increases Ion Leakage in *Arabidopsis* and N. benthamiana

To quantify the effect of Gβγ expression on PCD-related responses, we assessed the induction of ROS production by flg22, which serves as an indicator, and ion leakage as a measure of the severity of cell wall degradation. We overexpressed AGB1 and AGG1 cloned in pBYR2eFa in *Arabidopsis* and *N. benthamiana* leaves before subjecting them to flg22 treatment and compared the results to a similar treatment in plants overexpressing GFP. As expected, the plants expressing AGB1/AGG1 displayed a marked increase in flg22-induced ROS production compared to control leaves expressing GFP (Figure 2A,B).

Cell death leads to the loss of plasma membrane integrity, allowing ions and small molecules to leak from cells and alter the ionic composition of their surroundings [54]. Thus, conductivity values provide an indirect measure of the severity of cell death in a tissue. It is well known that Agrobacterium-mediated transient expression is not consistent across infiltrated tissues; therefore, we took advantage of this variability and used the HiBiT tag fused to AGB1 to test whether there was a correlation between AGB1/AGG1 protein levels and ion leakage. We infiltrated *Arabidopsis* and *N. benthamiana* leaves with pBYR2eFa-HiBiT-AGB1/pBYR2eFa-AGG1 constructs. Subsequently, we determined both the conductivity and the relative AGB1/AGG1 protein levels in approximately one hundred leaf discs sourced from the infiltrated tissues. Our results showed a positive correlation between the relative amounts of AGB1 protein and ion leakage (Figure 2C,D). These results further support a causative enhancing effect of G proteins on cell death progression. Note that this effect was consistently observed in leaf tissues of both *Arabidopsis* and *N. benthamiana*, highlighting the evolutionary conservation of the G protein-mediated enhancement of cell death.

### 3.3. Gβγ and XLG2 Play Interdependent Roles in the Cell Death Progression

Studies on *Arabidopsis* G protein knockout mutants have shown that AGB1/AGG1 and AGB1/AGG2 Gβγ dimers, as well as XLG2 and XLG3, play a cooperative role in plant immunity, while GPA1 and XLG1 do not contribute to this process [35]. In agreement with this data, overexpression of AGB1/AGG1 resulted in the enhanced cell death in Col-0, *gpa1-3*, and *xlg1-1* mutants, but not in *xlg2 xlg3* double mutants (Figure 3A), indicating that functional XLG2 or XLG3 are necessary for the response. We then tested whether overexpression of XLG2 leads to symptoms similar to those observed in AGB1/AGG1 overexpression. In *N. benthamiana* and *Arabidopsis*, overexpression of AGB1/AGG1 or XLG2 was associated with severe necrosis, while GFP overexpression displayed only incipient necrosis (Figure 3B,C). Quantitative analysis on *Arabidopsis* leaves confirmed that the area affected by necrosis was significantly larger in leaves overexpressing XLG2 or AGB1/AGG1 than in controls overexpressing GFP (Figure 3D).

Next, we investigated whether the cell death phenotype caused by AGB1/AGG1 overexpression is dependent on XLG2 and vice versa and whether the cell death phenotype caused by XLG2 overexpression is dependent on AGB1/AGG1. For this purpose, we expressed AGB1/AGG1 in *xlg2-1* mutant and, conversely, overexpressed XLG2 in an *agb1-2* mutant. Interestingly, the enhanced necrosis caused by AGB1/AGG1 overexpression in Col-0 was not observed in *xlg2-1* leaves (Figure 3E), while the enhanced necrosis caused by XLG2 overexpression in Col-0 was absent in *agb1-2* mutant (Figure 3F), indicating that both AGB1/AGG1 and XLG2 functional subunits are necessary for PCD progression enhancement. In further support of the interdependence of XLG2 and AGB1/AGG1 functions, we examined their effect using *Arabidopsis* PR1 promoter fused to the firefly luciferase gene. This construct was co-expressed with either GFP, AGB1/AGG1, XLG2, or XLG2/AGB1/AGG1 in *N. benthamiana* leaves. Our results showed stronger luciferase luminescence upon co-expression with AGB1/AGG1 or XLG2, but not GFP (Figure 3G). The absence of an additive effect upon simultaneous overexpression of AGB1/AGG1 and XLG2 (Figure 3G) confirmed that these subunits function together in a coordinated manner to regulate PR1 gene expression, implying that they are involved in the same regulatory pathway.

PR genes can be used as markers for cell death since their expression is often increased in cells undergoing cell death in response to pathogen infection. We quantified *PR1*, *PR2*, *PR3*, *PR4*, and *PR5* transcript levels upon overexpression of AGB1/AGG1 or XLG2 in Col-0, *xlg2 xlg3*, and *agb1-2* mutants and compared them to the levels observed upon overexpression of GFP as control. Overexpression of AGB1/AGG1 in Col-0 resulted in a significant increase in *PR1*, *PR2*, *PR3*, *PR4*, and *PR5* transcript levels compared to the levels measured in the GFP controls (Figure 4A–E), while XLG2 overexpression increased transcript levels in all genes except for *PR3*. It is notable that overexpression of AGB1/AGG1 did not induce the expression of *PR* genes in *xlg2 xlg3* mutants, and similarly, overexpression of XLG2 did not induce the expression of *PR2*, *PR3*, *PR4*, and *PR5* in *agb1-2* mutant (Figure 4A–E). The exception was the *PR1* gene, whose levels were strongly induced in *agb1-2* mutant via overexpression of both GFP and XLG2, suggesting that this increase in expression is not specific to XLG2 overexpression. Overall, these results indicate that Gβγ and XLG2 function interdependently but can also control specific responses.

### 3.4. G Proteins Mediate Cell Death Progression in a Salicylic Acid-Dependent Manner

PCD is a complex process that involves multiple pathways and regulatory mechanisms. Typically, the sequence of events leading to cell death begins with specific receptor-like kinases perceiving outside cues and initiating signal transduction pathways, such as SA-mediated pathways. Increases in SA levels lead to the activation of the EDS1/PAD4 complex, which plays a key role in the expression of PR genes. This can ultimately result in the initiation of cell degenerative processes [55,56].

In *Arabidopsis*, the receptor-like kinase BIR1 is a negative regulator of the plant defense response and cell death. *bir1* mutants display severe autoimmune phenotypes characterized by constitutively activated defense responses, generalized cell death, and temperature-dependent seedling lethality [57]. The cell death exhibited by the *bir1-1* mutant is partially suppressed by mutations in the *SOBIR1*, *AGB1*, *AGG1*/*AGG2*, *XLG2*, and *PAD4* genes, indicating that these proteins play positive roles in the cell death response [35,40,41]. *SOBIR1* and *AGB1* appear to act independently of *PAD4* in *bir1-1*-induced PCD [40,57]. Furthermore, they each contribute independently to *Arabidopsis* immunity [58]. We observed that when AGB1/AGG1 were overexpressed in *sobir1-12* plants, the level of cell death was similar to that observed in wild-type Col-0 (Figure 5), indicating that SOBIR1 acts either independently or upstream of AGB1/AGG1 in promoting cell death. On the other hand, overexpression of AGB1/AGG1 in SA-deficient plants expressing bacterial *NahG* and *pad4-1* mutants failed to elicit the levels of necrosis observed in Col-0 (Figure 5), suggesting that AGB1/AGG1-mediated cell death progression is dependent on SA signaling and PAD4 in particular. Alternatively, it is also possible that the experimental treatment using the pBYR2eFa vector failed to initiate cell death in the SA-impaired mutants, and therefore, the Gβγ-mediated enhancement of PCD progression cannot be observed. Overall, these results suggest that the cell death enhancing effect of AGB1/AGG1 may be regulated by multiple genes and signaling pathways.

## 4. Discussion

Programmed cell death is a critical part of plant immunity, as it restricts the spread of pathogens and allows the plant to protect uninfected areas [1]. Plants have evolved a complex network of signaling pathways and molecular interactions with pathogens to regulate cell death and balance tissue loss with overall plant fitness. Cell death is tightly regulated by a variety of genes and signaling molecules, including reactive oxygen species, calcium ions, and plant hormones such as salicylic acid, ethylene, and jasmonic acid [2]. The process of PCD can be divided into three distinct phases: initiation, progression, and termination [5]. It is reasonable to suggest that each of these phases is regulated via distinct molecular mechanisms, although these mechanisms have not been fully established.

Using two distinct expression systems—pCambia1300, which facilitates the high transient expression of the genes of interest and pBYR2eFa, containing not only the gene of interest but also the gene for the geminiviral Rep protein known to induce cell death—we demonstrated that the simultaneous overexpression of Gβ and Gγ subunits or the sole XLG2 subunit of heterotrimeric G proteins significantly enhanced PCD, but did not actually initiate it. Our findings were further supported by the absence of PCD enhancement in SA-deficient or SA-insensitive mutants. This suggests that G proteins might act upstream of the SA signaling pathway in mediating cell death, or alternatively, the geminiviral Rep protein might require fully functional SA signaling to initiate cell death. While prior research indicates that G proteins operate mostly independently of SA in plant defense responses [22], our study did not definitively distinguish between these two possibilities. Importantly, our overexpression strategy exposed a positive regulatory mechanism for PCD progression in tomato, where Gβ knockout leads to pronounced autoimmune responses, precluding the assessment of positive effects through conventional loss-of-function mutations. Tomato is one of four diverse species where knockout mutation of Gβ led to an autoimmune response [19]. This observation suggests that G proteins likely play roles in two distinct mechanisms regulating different aspects of cell death in plants. One of these mechanisms enhances defense responses to pathogens and positively regulates PCD progression, whereas the other appears to be intricately associated with the initiation phase of PCD. Given that both mechanisms are evident in tomato, it is reasonable to hypothesize that such regulatory patterns may be common across a broader spectrum of plant species, while *Arabidopsis* likely lost the mechanism initiating PCD in the course of evolution.

While the details of these two mechanisms are subjects for future research, some speculative insights can be derived. One plausible hypothesis can be made within the framework of the “guard hypothesis” or “guard model” describing plant–pathogen interactions. According to the “guard hypothesis”, plants possess specific proteins termed “guardees”, which serve as targets for pathogen effectors. Upon modification or degradation of these guard proteins by pathogen effectors, the plant cell detects their altered state or absence and initiates defense responses, including PCD, aimed at limiting pathogen proliferation and dissemination [59,60]. It is now well-recognized that G proteins are important positive players in plant defense. Consequently, they might become targets for pathogen effectors, which typically manipulate critical proteins to subdue host immunity [60]. In this context, the constitutively active defense responses may arise from the initiating of the effector-triggered immunity (ETI) response. This activation could be attributed to the absence of Gβγ dimers which might be recognized and guarded by a yet-to-be-identified nucleotide-binding leucine-rich repeat (NLR) receptor. Arguably, such a receptor seems to be lost in *Arabidopsis*, potentially explaining its distinct managing of the PCD initiation phase.

Our findings indicate that while the Gβγ dimer can enhance and/or maintain cell death progression, it does not serve as the initiating factor. This observation aligns with the previous research, which points out the role of G proteins in modulating defense responses rather than directly triggering them [61,62]. The enhancement of the cell death response mediated by AGB1/AGG1 was found to be XLG2-dependent but did not involve GPA1 or XLG1. This is consistent with earlier studies that highlight a functional association between XLG2 and Gβγ-mediated defense signaling [10,33,34,35]. Our data also indicated that the XLG2-dependent enhancement of cell death required the presence of AGB1/AGG1, implying that these signaling components participate in a shared pathway. However, variations observed in PR1 gene expression suggest that these subunits might also have minor distinct roles.

In summary, our study reconciles the apparent discrepancies observed between Gβ-knockout mutants in *Arabidopsis* and other species and improves the current understanding of the role of G proteins in plant cell death. Our results emphasize the importance of studying G protein signaling in multiple species to gain a comprehensive understanding of their role. Future studies should aim to investigate Gβγ-downstream targets in the programmed cell death signaling pathways to gain a deeper understanding of the molecular mechanisms underlying this process.

## Figures and Tables

**Figure 1 genes-15-00115-f001:**
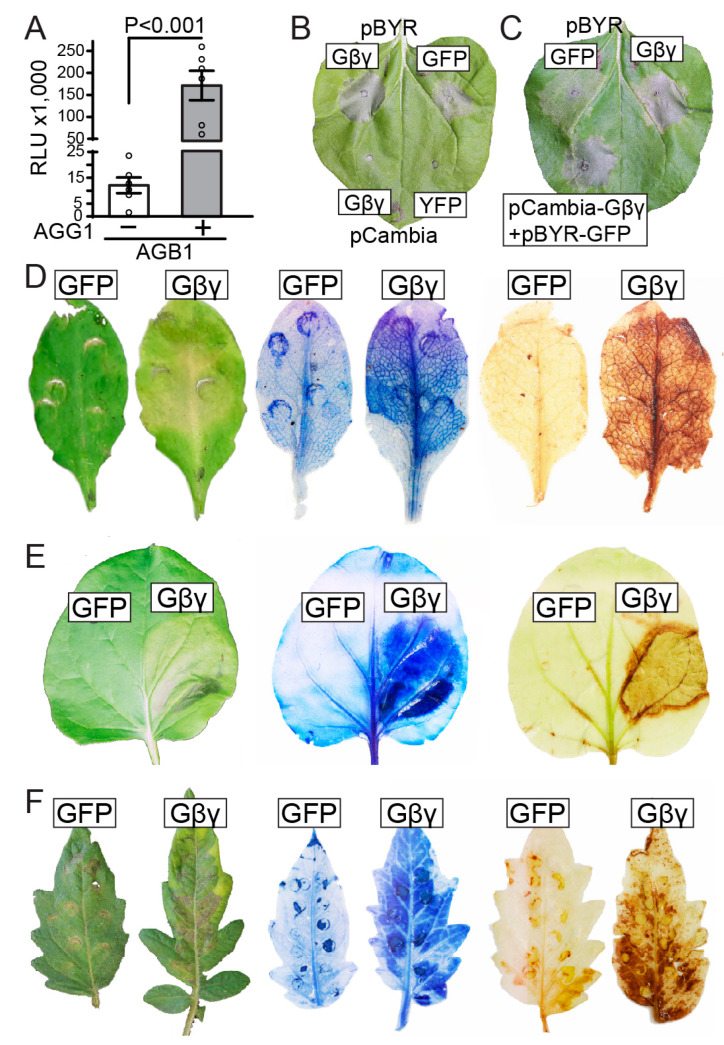
Gβγ overexpression enhanced, but did not initiate PCD in leaves of *Arabidopsis*, *Nicotiana benthamiana* and tomato. (**A**) Stable expression of AGB1 requires AGG1. The *AGB1* cDNA was tagged with HiBiT in pCambia1300 vector and transiently expressed in *N. benthamiana* leaves with or without co-expression with AGG1. The graph shows mean values of relative light units obtained from HiBiT assays based on 6 measurements. (**B**) and (**C**) The *GFP*, *AGB1*, and *AGG1* coding regions were cloned into the pBYR2eFa and pCambia1300 vectors and co-expressed in *N. benthamiana* leaves using *Agrobacterium* infiltration. pBYR2e-GFP showing mild necrosis and pCambia1300-YFP showing no necrosis were used as negative controls. (**D**,**E**) Necrotic lesions (trypan blue staining and DAB staining) in (**D**) *Arabidopsis* and (**E**) *N. benthamiana* leaves infiltrated with pBYR2e-GFP or pBYR2eFa-AGB1/pBYR2eFa-AGG1. (**F**) Necrotic lesions (trypan blue staining and DAB staining) in tomato leaves infiltrated with pBYR2e-GFP or pBYR2eFa-SlGB1/pBYR2eFa-SlGGA1 to co-express the tomato Gβ and Gγ subunits. The necrotic lesions (trypan blue staining and DAB staining) were repeated on at least three different leaves for each species with similar outcomes.

**Figure 2 genes-15-00115-f002:**
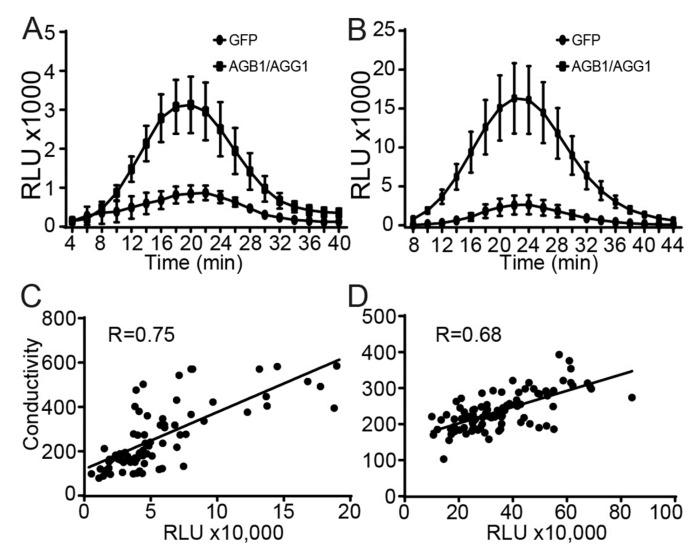
Overexpression of Gβγ enhances flg22-induced ROS production and increases ion leakage in *Arabidopsis* and *N. benthamiana*. (**A**,**B**) flg22-induced ROS production in (**A**) *Arabidopsis* and (**B**) *N. benthamiana* leaves infiltrated with pBYR2e-GFP or pBYR2eFa-AGB1/pBYR2eFa-AGG1. Leaf discs were collected three days after infiltration and incubated in sterile water for 24 h in darkness. Discs were then treated with 1 μM flg22 and subjected to a luminol-based assay. Graphs show the average values of light emission with standard errors (*n* = 12). (**C**,**D**) Correlation between ion leakage and AGB1 protein levels in (**C**) *Arabidopsis* and (**D**) *N. benthamiana* leaves infiltrated with pBYR2eFa-HiBiT-AGB1/pBYR2eFa-AGG1. Three days after infiltration, leaf discs were placed in 200 µL of distilled sterile water. The conductivity of each sample was measured before replacing the water with 30 μL of Nano-Glo^®^ HiBiT reagent buffer to measure luminescence. Graphs show a dot plot for each leaf disc with the corresponding correlation coefficient.

**Figure 3 genes-15-00115-f003:**
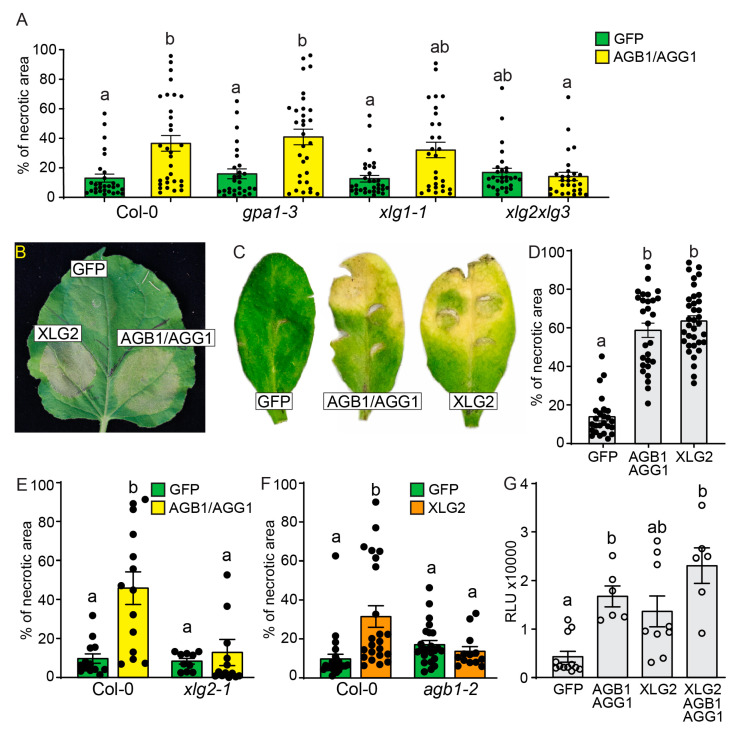
AGB1/AGG1 and XLG2 enhance cell death interdependently. (**A**) Overexpression of *AGB1*/*AGG1* or *GFP* in Col-0, *gpa1-3*, *xlg1-1*, and *xlg2 xlg3* mutants. The graph shows the average percentage of the necrotic lesion area (*n* > 28). (**B**,**C**) Leaves of (**B**) *N. benthamiana* and (**C**) *Arabidopsis* infiltrated with pBYR2e-GFP, pBYR2eFa-XLG2, or pBYR2eFa-AGB1/pBYR2eFa-AGG1. (**D**) Quantification of the necrotic area in *Arabidopsis* leaves infiltrated with pBYR2e-GFP, pBYR2eFa-XLG2, or pBYR2eFa-AGB1/pBYR2eFa-AGG1. (*n* > 25). (**E**) Quantification of the necrotic area in *Arabidopsis* Col-0 and *xlg2-1* mutant leaves infiltrated with pBYR2e-GFP or pBYR2eFa-AGB1/pBYR2eFa-AGG1 (*n* > 10). (**F**) Quantification of the necrotic area in *Arabidopsis* Col-0 and *agb1-2* mutant leaves infiltrated with pBYR2e-GFP or pBYR2eFa-XLG2 (*n* > 13). (**G**) Activation of the *PR1* promoter via G protein overexpression. The *PR1* promoter was cloned upstream of the firefly luciferase cDNA and co-expressed along with either pBYR2e-GFP, pBYR2eFa-AGB1/pBYR2eFa-AGG1, pBYR2eFa-XLG2, or pBYR2eFa-AGB1/ pBYR2eFa-AGG1/pBYR2eFa-XLG2 in *N. benthamiana* leaves. Luciferase activity was measured three days after infiltration (*n* > 6). In all graphs, bars show the mean ± SEM, with circles representing individual values. Compact letter display (CDL), “a” and “b”, indicates groups with statistically significant differences analyzed with a non-parametric Kruskal–Wallis test, followed by Dunn’s test for multiple comparisons (*p* < 0.05). Graph generation and statistical analyses were performed with GraphPad Prism 9.5.1.733 software.

**Figure 4 genes-15-00115-f004:**
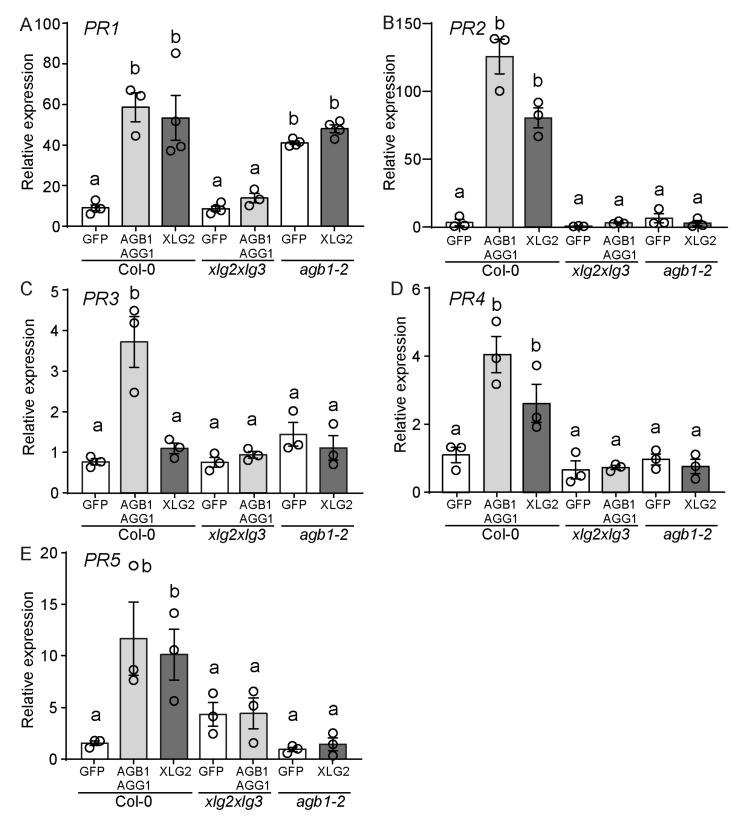
Overexpression of AGB1 and AGG1 activates expression of pathogenesis-related (*PR)* genes. RT-qPCR quantification of transcript levels of *PR* genes (**A**) *PR1*, (**B**) *PR2*, (**C**) *PR3*, (**D**) *PR4*, and (**E**) *PR5*. Wild-type (Col-0) plants, *xlg2 xlg3* and *agb1-2* mutants were infiltrated with pBYR2e-GFP, pBYR2eFa-AGB1/pBYR2eFa-AGG1, or pBYR2eFa-XLG2, respectively. RNA was extracted three days after infiltration. The *SAND* (AT2G28390) expression levels were used for normalization purposes. Relative expression is shown as the mean ± SEM, *n* = 3, with circles representing individual values. Compact letter display (CLD), “a” and “b”, was used to signify statistically significant differences identified via one-way ANOVA with Tukey’s test, *p* < 0.05.

**Figure 5 genes-15-00115-f005:**
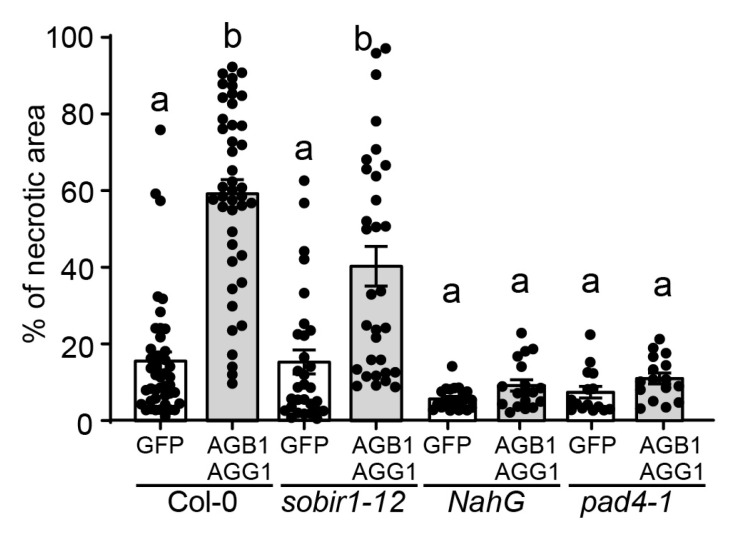
G proteins mediate cell death in a SA-dependent manner. Quantification of the necrotic area via transient expression of pBYR2e-GFP or pBYR2eFa-AGB1/pBYR2eFa-AGG1 in WT Col-0 plants, *sobir1-12*; *pad4-1* mutants and transgenic *NahG Arabidopsis* leaves. Bars show the mean ± SEM (*n* > 10), with circles representing individual values. Compact letter display (CDL), “a” and “b”, indicates groups with statistically significant differences analyzed with a non-parametric Kruskal–Wallis test, followed by Dunn’s test for multiple comparisons (*p* < 0.05). Graph generation and statistical analyses were performed with GraphPad Prism 9.5.1.733 software.

## Data Availability

The data presented in this study are available on request from the corresponding author. The data are not publicly available due to privacy.

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
