# Peer review of "Conserved Role of Heterotrimeric G Proteins in Plant Defense and Cell Death Progression"

_genes, 2024, doi:10.3390/genes15010115_

Round 1

Reviewer 1 Report

Comments and Suggestions for Authors

Dear respected colleagues,

Upon reviewing your paper, I found the content and scientific theme to be intriguing. In this paper, Karimian et al. critically evaluate the critical role of G proteins in plant defense and programmed cell death.

The manuscript style of writing, discussion, and management of scientific content are perfect. Such papers will improve the quality of literature and pave the way for further large-scale studies in this field.

However, the respected authors should address several comments to increase the quality of the manuscript. Please consider the following comments and amend the manuscript content accordingly:

1. Please ensure that all cited references include their respective DOI identifiers, which are crucial for facilitating access to the sources and enhancing the credibility of the paper.

2. It would be beneficial to provide a list of abbreviations for all summarized terms, as this will aid in improving the readability and comprehension of the manuscript.

3. Kindly verify that the reference list does not include problematic papers, such as retracted papers or similar records, as citing these could impact the integrity of the research. Ensuring the reliability of the references is essential for the overall quality of the paper.

4. Consider enhancing the visibility of the work in the literature by incorporating a flowchart to summarize the methodology. This visual aid can effectively communicate the research approach and findings to the readers.

5. To further support the study, please include a complete map of the expression vector used in this research. This additional information will contribute to a comprehensive understanding of the experimental procedures.

6. It would be valuable to mention the names of all tools used for graphical representations of bar plots and statistical analysis, thereby providing transparency and enabling other researchers to replicate and build upon the findings.

7. Please provide a clear explanation of the statistical methods and p-value levels used to analyze the experimental data. Ensuring transparency in the statistical analysis is essential for the robustness of the research findings and the interpretation of the results.

8- Lines 308-316: Please add relevant references to this section of discussion.

9- Lines 147-152: Please provide more details about the discussed methodology in this section.

10- Please supplement the electrophoresis gels of the cDNA amplification assay. 

11- Please supplement the melting curves of RT-qPCR assays for the studied genes. 

12- The infiltration process alone can induce necrosis symptoms in the treated leaves. Can the respected authors provide the images of their control groups? 

13- Different transcription factors affect the expression of G proteins in Arabidopsis, tobacco, and tomato plants. For example, MYB62 is one of the most important TFs involved in G protein actions within plant cells. I highly recommend that the authors discuss the roles of critical transcription factors linked to programmed cell death and G protein signaling pathways in this study. You can also add some bioinformatics analyses to your work to explain how TFs can regulate the observed molecular responses of the studied plants after infiltration process with the evaluated constructs. Such views are rarely discussed in the literature, and adding this section to their paper will significantly increase the visibility of this paper among other relevant works published in this field. You can use MEME-SUITE, TBtools, and other relevant bioinformatics pipelines to provide the data connected to this section.

14- Since different plants were evaluated in this study, the authors must provide a view of how the amino acid and nucleotide sequences of G proteins of these plants vary. You can construct a multiple alignment file, compare the sequences to each other and homologous ones in other plants, and build a phylogenetic tree to evaluate the differences between the sequences of G proteins in the studied plants.

15- In the Figure 3 caption, please mention which statistical software or R package was used to construct the statistical bar plots.

16- How do the respected authors confirm that the constructed vectors were successfully transferred into the studied plants? May I kindly ask you to provide the outputs of protein assays for this case?

17- The GFP reporter gene mainly produces a green color in treated plant tissues while in this study the reported color for the GFP reporter gene is blue. May I kindly ask you to clarify this case, too?

18- Lines 323-325: Please provide a reference for the discussed text. If this claim is a hypothesis, the authors can use in silico assays using available tools to confirm their hypothesis and prove the association between G protein activity and SA signaling pathway.

19- Please add a concluding remark to the end of your discussion and highlight the weaknesses and strengths of this study.

However, the respected authors must provide a clear answer to all given queries. The rest of the paper's data and style have no problem and the respected authors successfully evaluated their hypothesis and reach an acceptable and powerful conclusion about the studied treatments. At this step, I have no further comments and my recommendation for this paper is “major revision”.

Best regards,

Author Response

We would like to thank the reviewer for taking the time to review this manuscript and useful suggestions which helped to improve the manuscript. Please find the detailed responses below. 

Comment 1. Please ensure that all cited references include their respective DOI identifiers, which are crucial for facilitating access to the sources and enhancing the credibility of the paper.

Response 1 The citation list was corrected accordingly.

Comment 2. It would be beneficial to provide a list of abbreviations for all summarized terms, as this will aid in improving the readability and comprehension of the manuscript.

Response 2 The list of abbreviations was added.

Comment 3. Kindly verify that the reference list does not include problematic papers, such as retracted papers or similar records, as citing these could impact the integrity of the research. Ensuring the reliability of the references is essential for the overall quality of the paper.

Response 3 The citations were checked, and no problematic papers were found.

Comment 4. Consider enhancing the visibility of the work in the literature by incorporating a flowchart to summarize the methodology. This visual aid can effectively communicate the research approach and findings to the readers.

Response 4 All methods in this work were previously published and appropriate citations are provided.

Comment 5. To further support the study, please include a complete map of the expression vector used in this research. This additional information will contribute to a comprehensive understanding of the experimental procedures.

Response 5 Images of complete vectors maps were added to suppl material.

Comment 6. It would be valuable to mention the names of all tools used for graphical representations of bar plots and statistical analysis, thereby providing transparency and enabling other researchers to replicate and build upon the findings.

Response 6 The software name and statistical tools were added.

Comment 7. Please provide a clear explanation of the statistical methods and p-value levels used to analyze the experimental data. Ensuring transparency in the statistical analysis is essential for the robustness of the research findings and the interpretation of the results.

Response 7 Statistical methods were added in Methods section.

Comment 8- Lines 308-316: Please add relevant references to this section of discussion.

Response 8 The references were added.

Comment 9- Lines 147-152: Please provide more details about the discussed methodology in this section.

Response 9 More details were added.

Comment 10- Please supplement the electrophoresis gels of the cDNA amplification assay. 

Response 10 cDNA amplification was performed according to manufacturer’s instructions and was not assayed on gel. To ensure the integrity and quality of cDNA we used reference gene SAND as mentioned in the manuscript. 

Comment 11- Please supplement the melting curves of RT-qPCR assays for the studied genes.

Response 11 Unfortunately, the experiments were performed three years ago and original files with melting curves data are no longer available.

Comment 12- The infiltration process alone can induce necrosis symptoms in the treated leaves. Can the respected authors provide the images of their control groups? 

Response 12 The respected reviewer is correct about syringe infiltration causing the mechanical damage. For that reason, we used leaves infiltrated with GFP-vectors as control for syringe induced damage, agrobacterial growth effects and high levels of a foreign protein. The corresponding images and necrosis evaluations are shown in all experiments next to proteins of interest (Gβγ and XLG2).

Comment 13- Different transcription factors affect the expression of G proteins in Arabidopsis, tobacco, and tomato plants. For example, MYB62 is one of the most important TFs involved in G protein actions within plant cells. I highly recommend that the authors discuss the roles of critical transcription factors linked to programmed cell death and G protein signaling pathways in this study. You can also add some bioinformatics analyses to your work to explain how TFs can regulate the observed molecular responses of the studied plants after infiltration process with the evaluated constructs. Such views are rarely discussed in the literature, and adding this section to their paper will significantly increase the visibility of this paper among other relevant works published in this field. You can use MEME-SUITE, TBtools, and other relevant bioinformatics pipelines to provide the data connected to this section.

Response 13 We agree with the reviewer regarding the importance of researching further into the mechanisms regulating G proteins and downstream signalling cascades. However, discussing these potential mechanisms without supporting experimental evidence on our part would lead to excessive speculation. Given that this article primarily reports experimental results, we would like to avoid making too many ad hoc hypotheses.

Comment 14- Since different plants were evaluated in this study, the authors must provide a view of how the amino acid and nucleotide sequences of G proteins of these plants vary. You can construct a multiple alignment file, compare the sequences to each other and homologous ones in other plants, and build a phylogenetic tree to evaluate the differences between the sequences of G proteins in the studied plants.

Response 14 We did indeed utilize Gβ and Gγ subunits from Arabidopsis and tomato. However, all four genes and respective proteins have been comprehensively described in previous studies. Additionally, the construction of the phylogenetic tree, encompassing various plant species, was previously reported and is not directly relevant to the primary objective of our research.   

Comment 15- In the Figure 3 caption, please mention which statistical software or R package was used to construct the statistical bar plots.

Response 15 The required information was added.

Comment 16- How do the respected authors confirm that the constructed vectors were successfully transferred into the studied plants? May I kindly ask you to provide the outputs of protein assays for this case?

Response 16 The successful transfer of the vectors is confirmed by positive results of HiBiT-tag assay as it is the equivalent of protein assay, for instance on Fig. 2C and D. Luminescence was used as evaluation of the tag presence (similar to commonly used, His6-tag, GST-tag or Flag-tag) all quantifiable results presented as relative light units, RLUs. In other experiments, symptomatic necrotic lesions were evaluated signifying that the vectors were transferred, and the proteins were expressed.

Comment 17- The GFP reporter gene mainly produces a green color in treated plant tissues while in this study the reported color for the GFP reporter gene is blue. May I kindly ask you to clarify this case, too?

Response 17 We apologise but we did not understand the question. In this work, we did not assay GFP fluorescence. The vector with GFP gene was merely used as an empty vector to provide a control showing that infiltration without AGB1/AGG1, the PCD symptoms were minor or absent. This is a commonly used practice along with empty vector controls. If the respected reviewer refers to Fig. 1D, E,F, the blue color is the trypan blue staining, showing the areas with necrotic cells.

Comment 18- Lines 323-325: Please provide a reference for the discussed text. If this claim is a hypothesis, the authors can use in silico assays using available tools to confirm their hypothesis and prove the association between G protein activity and SA signaling pathway.

Response 18 This text provides two possible explanations of our observed results and is not a claim. As mentioned in the next sentence, we could not rule out which of the two is the correct explanation. In silico assays in this case have to be supported by experimental work and could not provide a definitive proof for either scenario. Although it is an interesting question regarding SA-G proteins interplay, it is not major for our main conclusion, therefore  we will research into it in a future project.

Comment 19- Please add a concluding remark to the end of your discussion and highlight the weaknesses and strengths of this study.

Response 19 The strengths and weaknesses of the experiments and hypotheses are addressed within the manuscript. The concluding summary emphasizes the primary outcomes of the results, adhering to the conventional style of specialized scientific articles.

Reviewer 2 Report

Comments and Suggestions for Authors

In this investigation, Karimian and collaborators utilized a transient overexpression method to probe the functions of Gβγ in programmed cell death (PCD) across Arabidopsis, tomato, and Nicotiana benthamiana. The authors illustrated that overexpression of these G proteins consistently heightened PCD in all three species. They also presented evidence suggesting that the Gαβγ subunits, AGB1, AGG1, and XLG2/3, may function in the same pathway to regulate the progression of PCD. The contrasting phenotypes observed in Arabidopsis mutants compared to tomato mutants in G proteins prompted the authors to propose a potential role of G proteins as guardians, with a mechanism that might be lost in Arabidopsis. This is indeed intriguing. The paper is commendably written and executed, with all conclusions robustly supported by the reported data. While I have no significant concerns with this work, I offer a minor suggestion: including quantitative analysis and repeated numbers in Fig 1B-F would enhance the comprehensiveness of the presentation, moving beyond a representative figure.

Author Response

We would like to thank the reviewer for taking the time to review this manuscript and useful suggestions which helped to improve the manuscript. Please find the detailed responses below.

Comment 1 including quantitative analysis and repeated numbers in Fig 1B-F would enhance the comprehensiveness of the presentation, moving beyond a representative figure.

Response 1 The number of repeats was added to the figure legend. We believe that the quantitative analysis of staining (both trypan blue and DAB) would be unnecessary as main aim here was to show the qualitative difference. Furthermore, quantification in this case might be even misleading, as it might  suggest a magnitude of PCD enhancement caused by G proteins, while correlation between enhancement effect and intensity of staining is rarely linear. Relative quantification was done in next section for ROS and ion leakage, see fig. 2

Round 2

Reviewer 1 Report

Comments and Suggestions for Authors

Dear authors, 

Thank you very much for your revision and re-submitting the paper to this journal. However, based on the provided answers in the authors’ reply form to the reviewer(s) commentsÙˆ the respected authors did not revise the manuscript based on the given comments. Simple and optional comments to improve the content of this paper and its visibility in the literature have been provided for the respected authors, and there were not appropriate amendments within the paper. However:

- I asked the authors to provide a list of abbreviations for all summarized terms and it was not added to the paper.

-I asked the authors to provide extra information in the supplementary files but the comment was not addressed thoroughly. 

- Within the paper, the authors discussed the role of specific transcription factor in this assay, however, its interaction with G proteins signaling cascades was not deeply discussed. 

- The respected authors considered the reviewer comments as ad hoc hypotheses while in any scientific paper amending the manuscript based on the reviewer(s) comments is necessary to prepare the manuscript data and content for publishing.

Please note that reviewers and authors work together to improve the quality of scientific papers submitted to academic journals to generally increase the quality of the available literature and pave the way for conducting further studies in specific lines related to the reviewed papers.

- It was not clear why the respected authors considered different plants in this study and did not discuss the differences in observed results in these plants.  The control groups for all studied plants were missed in this study and the respected authors have not compared their results with the control groups.

- It was also not clear in which step of plant growth infiltration assay was conducted.

- The respected authors mentioned that they used Tukey’s multiple comparison for statistical analysis while it was not clear how many repeats and control groups were considered for this study. The assigned words- based on the Tukey test on the top of each bar plot represented in Figures 3 and 4 are not matched with the size of bar plots and given numbers on the Y axis.

There are differences between some bar plots illustrated in these charts while the respected authors ignored the differences and did not statistically evaluate the observed differences. Another point that should be addressed is that the respected authors did not mention why they considered Tukey’s multiple comparisons for their analysis.

Although this method is widely used to analyze experimental data, it has significant sensitivity to outliers particularly when the sample sizes are too small. Before using this statistical method, the assumption of the normality of data selected for statistical analysis should be evaluated. As I checked the methodology of the revised version, I could not find how the respected authors managed their repeats and treatments and how they proved the assumption of the normality of the analyzed data before applying Tukey’s multiple comparisons.

However, it can be too long to discuss the error of the applied statistical methodology in this paper and the respected authors should first check their data consistency with Tukey’s multiple comparison test and then use this statistical methodology to evaluate their data.

Altogether, the manuscript revision was not satisfying and the manuscript content has not been critically amended based on the given comments. The manuscript topic and goal are intriguing and mind-catching, however, at this step, my recommendation for this paper is “Rejection” due to significant errors in the applied statistical methods, the lack of appropriate comparison between control and treated groups, the lack of experimental design information and the number of repeats and treatments used in this study and ignoring the reviewer(s) comments to improve the scientific content of this paper. 

Author Response

Rev1

Dear authors, 

Thank you very much for your revision and re-submitting the paper to this journal. However, based on the provided answers in the authors’ reply form to the reviewer(s) commentsÙˆ the respected authors did not revise the manuscript based on the given comments. Simple and optional comments to improve the content of this paper and its visibility in the literature have been provided for the respected authors, and there were not appropriate amendments within the paper. However:

- I asked the authors to provide a list of abbreviations for all summarized terms and it was not added to the paper.

Response: We apologize for the oversight regarding the list of abbreviations. While we did create the list, we inadvertently failed to include it in the manuscript. Our submission followed to the journal's template, which lacks a designated section for abbreviations. We planned to include the list in the Supplementary Materials; however, it appears that we overlooked this in the final stages of revised file preparation. The updated Supplementary Materials now include the abbreviation list. If the reviewer identifies any missing terms in the list, please specify which ones we should incorporate.

-I asked the authors to provide extra information in the supplementary files but the comment was not addressed thoroughly. 

Response: We included additional information in our resubmission, although we acknowledge that some requested information is no longer available. We apologize for any inconvenience caused. We are committed to addressing this concern. To facilitate our response, we kindly request that the reviewer specify the type and extent of additional information required or any areas needing further clarification. We are more than willing to provide any missing details. However, we are uncertain about the term 'flowchart to summarize the methodology' as a visual aid. Could the reviewer please provide an example from published literature or accessible on the internet that we could use as a template? Thank you for your understanding, and we appreciate the opportunity to improve our paper.

- Within the paper, the authors discussed the role of specific transcription factor in this assay, however, its interaction with G proteins signaling cascades was not deeply discussed.

Response: We apologize for any confusion, but our paper did not include a discussion on transcription factors, and no transcription factors were assayed in our study. Could you please specify the section or location of concern in our manuscript? We appreciate your feedback and will address the issue promptly.

- The respected authors considered the reviewer comments as ad hoc hypotheses while in any scientific paper amending the manuscript based on the reviewer(s) comments is necessary to prepare the manuscript data and content for publishing.

Response: We sincerely apologize for any misunderstanding. In our previous communication, we did not intend to call the reviewer's comments as ad hoc hypotheses. We only mentioned that we tried to avoid introducing too many hypotheses into our work. It is important to clarify that we acknowledge and appreciate the reviewer's insights, particularly regarding the significance of further research. While we agree with the reviewer on the necessity for additional research in this area, it is essential to emphasize that our current manuscript is only an initial description of the observed phenomenon. Our aim was not to establish the underlying mechanism but to present our findings as a starting point for future exploration. We acknowledge that manuscripts can be modified in response to reviewer comments, but such modifications typically occur when authors are in agreement with the suggested changes. In this instance, we respectfully disagree with the reviewer's suggestion. We kindly request the reviewer to consider our viewpoint on this matter. We value the constructive feedback provided and are open to further discussions to ensure the clarity of our position.

Please note that reviewers and authors work together to improve the quality of scientific papers submitted to academic journals to generally increase the quality of the available literature and pave the way for conducting further studies in specific lines related to the reviewed papers.

Response: We completely agree with this statement. 

- It was not clear why the respected authors considered different plants in this study and did not discuss the differences in observed results in these plants.  The control groups for all studied plants were missed in this study and the respected authors have not compared their results with the control groups.

Response: As we previously wrote, our designated control groups consisted of plants infiltrated with vectors carrying GFP gene. These control groups were selected for comparison with plants infiltrated with G protein vectors. As we understand, the reviewer is not satisfied with this explanation. We sincerely appreciate the feedback and would be grateful if the reviewer could specify the requirements for the preferred type of control group(s). We are committed to addressing these concerns to ensure the clarity of our study.

- It was also not clear in which step of plant growth infiltration assay was conducted.

Response: line 114 “Leaves of five-week-old plants either Arabidopsis or N. benthamiana were infiltrated”

- The respected authors mentioned that they used Tukey’s multiple comparison for statistical analysis while it was not clear how many repeats and control groups were considered for this study. The assigned words- based on the Tukey test on the top of each bar plot represented in Figures 3 and 4 are not matched with the size of bar plots and given numbers on the Y axis.

Response: We apologise but we used ANOVA with recommended Tukey test for multiple comparisons, and it showed that the differences between the means were statistically significant with p<0.05. For that reason we designated the differences with different letters.

There are differences between some bar plots illustrated in these charts while the respected authors ignored the differences and did not statistically evaluate the observed differences.

Response: We apologize for any confusion. Could the respected reviewer kindly specify the particular plot or plots containing bars without accompanying statistical analysis. Additionally, it would be helpful if the reviewer could highlight the specific differences that we may have overlooked.

Another point that should be addressed is that the respected authors did not mention why they considered Tukey’s multiple comparisons for their analysis.

Response: We utilized the commercial software GraphPad Prism 9.5.1.733, which offers various options and recommends the most suitable one for the analysed dataset. In our case Tukey’s test for multiple comparisons was recommended by this software.

Although this method is widely used to analyze experimental data, it has significant sensitivity to outliers particularly when the sample sizes are too small. Before using this statistical method, the assumption of the normality of data selected for statistical analysis should be evaluated. As I checked the methodology of the revised version, I could not find how the respected authors managed their repeats and treatments and how they proved the assumption of the normality of the analyzed data before applying Tukey’s multiple comparisons. However, it can be too long to discuss the error of the applied statistical methodology in this paper and the respected authors should first check their data consistency with Tukey’s multiple comparison test and then use this statistical methodology to evaluate their data.

Response: We sincerely appreciate the reviewer for pointing to the importance of testing the obtained data for normality. Regrettably, we acknowledge that we overlooked this crucial step, and the use of ANOVA was not justified in our initial submission. In response to this valuable feedback, we conducted normality tests on all our data, revealing that several experimental datasets did not pass the test for normality. Consequently, we have revised our statistical approach and replaced ANOVA with the non-parametric Kruskal-Wallis test, followed by Dunn’s test for multiple pairwise comparisons. These adjustments have been incorporated into the revised manuscript. We are grateful for the reviewer's insightful suggestion, and these improvements enhance the validity and reliability of our statistical analyses.

Altogether, the manuscript revision was not satisfying and the manuscript content has not been critically amended based on the given comments. The manuscript topic and goal are intriguing and mind-catching, however, at this step, my recommendation for this paper is “Rejection” due to significant errors in the applied statistical methods, the lack of appropriate comparison between control and treated groups, the lack of experimental design information and the number of repeats and treatments used in this study and ignoring the reviewer(s) comments to improve the scientific content of this paper. 

Response: We appreciate your valuable feedback and apologize for any shortcomings in the revised version. We understand your concerns regarding the statistical methods, comparison between control and treated groups, and the lack of experimental design information. We acknowledge the importance of addressing these issues to enhance the scientific quality of our paper. To correct these shortcomings, we thoroughly re-evaluated our statistical methods, provided a more comprehensive comparison between control and treated groups, and ensured the inclusion of detailed experimental design information, including corrected number of repeats used. Your comments are invaluable to us, and we are committed to making the necessary improvements to meet the standards of the journal. We are open to other specific suggestions you may have to guide us in the right direction. We appreciate the opportunity to address these concerns and enhance the quality of our manuscript.

Round 3

Reviewer 1 Report

Comments and Suggestions for Authors

Dear authors, 

Thank you very much for the revised version. This paper will certainly receive a considerable number of citations, and such high-quality academic papers are required to be published to enhance the quality of the available literature. I have no further comments on this paper. The respected authors appropriately revised the manuscript content and corrected all statistical errors. The current draft is suitable for publication in this journal and my recommendation for this paper is "acceptance". 

Best regards, 

Rasouli. H